# Inhibition of Phenol from Entering into Condensed Freshwater by Activated Persulfate during Solar-Driven Seawater Desalination

**DOI:** 10.3390/molecules27217160

**Published:** 2022-10-23

**Authors:** Xiaojiao Zhou, Ningyao Tao, Wen Jin, Xingyuan Wang, Tuqiao Zhang, Miaomiao Ye

**Affiliations:** 1Zhejiang Key Laboratory of Drinking Water Safety and Distribution Technology, College of Civil Engineering and Architecture, Zhejiang University, Hangzhou 310058, China; 2Donghai Laboratory, Zhoushan 316021, China

**Keywords:** interfacial solar heating, desalination, volatile organic compounds, phenol, persulfate

## Abstract

Recently, solar-driven seawater desalination has received extensive attention since it can obtain considerable freshwater by accelerating water evaporation at the air–water interface through solar evaporators. However, the high air–water interface temperature can cause volatile organic compounds (VOCs) to enter condensed freshwater and result in water quality safety risk. In this work, an antioxidative solar evaporator, which was composed of MoS_2_ as the photothermal material, expandable polyethylene (EPE) foam as the insulation material, polytetrafluoroethylene (PTFE) plate as the corrosion resistant material, and fiberglass membrane (FB) as the seawater delivery material, was fabricated for the first time. The activated persulfate (PS) methods, including peroxymonosulfate (PMS) and peroxodisulfate (PDS), were applied to inhibit phenol from entering condensed freshwater during desalination. The distillation concentration ratio of phenol (R_D_) was reduced from 76.5% to 0% with the addition of sufficient PMS or PDS, which means that there was no phenol in condensed freshwater. It was found that the Cl^−^ is the main factor in activating PMS, while for PDS, light, and heat are the dominant. Compared with PDS, PMS can make full utilization of the light, heat, Cl^−^ at the evaporator’s surface, resulting in more effective inhibition of the phenol from entering condensed freshwater. Finally, though phenol was efficiently removed by the addition of PMS or PDS, the problem of the formation of the halogenated distillation by-products in condensed freshwater should be given more attention in the future.

## 1. Introduction

The limited availability of fresh water often limits social development. For example, the high salinity of seawater can cause problems, such as the scaling of pipelines and corrosion of equipment, making it more difficult to extract oil in the Middle East [1]. Desalination can use a wide range of seawater resources to alleviate water pressure and scarcity. Recently, solar-driven seawater desalination based on interfacial solar heating has received great attention [2,3,4]. In this new approach, photothermal materials or evaporators floating on the water surface absorb solar energy and accelerate the evaporation of seawater, then condense water vapor to obtain freshwater [5]. Up to now, the water evaporation rate is generally higher than 1.30 kg/m^2^·h under one sun solar irradiance, and the temperature of the air–water interface can reach as high as 40–80 °C [6,7,8]. The higher temperature of the air–water interface means a better water evaporation rate [9,10]. However, pollutants in seawater, especially volatile organic compounds (VOCs), are more likely to enter condensed freshwater, resulting in water quality safety risks [11,12]. VOCs, such as phenol and BTEX, have a negative impact on the human nervous system [13], respiratory system [14], and immune system [15], as well as increasing the risk of cancer [16]. Therefore, the inhibition of VOCs from entering condensed freshwater has recently become a research hotspot in solar-driven seawater desalination.

At present, the main methods for inhibiting VOCs from entering into condensed freshwater are chemical oxidation processes, such as adsorption [17], photocatalysis [18], photo-Fenton [19], and activated persulfate [20]. For example, Liu et al. reported inhibition of styrene entry into freshwater by the adsorption process using Ag/UiO-66 nanocomposites both as the photothermal material and adsorbent [21]. Chen et al. reported the removal of phenol by the reaction of adsorption and photocatalysis under solar illumination via AC-TiO_2_/foam evaporator [22]. However, both adsorption and photocatalysis occur in the photothermal material layer with low removal efficiency because of the short contact time between VOCs and photothermal materials. Liang et al. reported the effective removal of VOCs through the Fenton reaction [23]. However, the low pH required for the reaction limits its large-scale use. It is therefore necessary to explore other more effective processes for the inhibition of VOCs from entering into condensed freshwater.

Sulfate radicals (SO_4_^•−^) based advanced oxidation processes are one of the effective methods to remove organic pollutants [24,25]. The activation of PS by light, heat, microwave, catalyst, etc. has been widely studied in the past several decades [26,27,28]. During solar evaporation, the surface of the evaporator has a high temperate temperature of 40–80 °C and a strong light intensity of 1 kW/m^2^, as well as a high dose of photothermal materials and halogen ions. It is reasonable to think that PS can be activated by heat, light, photothermal materials, and halogen ions at the same time. Therefore, VOCs can be inhibited from entering condensed freshwater by adding PS into the seawater during solar-driven seawater desalination.

In this work, an antioxidative evaporator was fabricated and applied for solar-driven seawater desalination for the first time. The evaporator was composed of MoS_2_ as the photothermal material, expandable polyethylene (EPE) foam as the insulation material, polytetrafluoroethylene (PTFE) plate as the corrosion-resistant material, and fiberglass membrane (FB) as the seawater delivery material. Phenol was selected as a typical VOC. The PMS and PDS were added to the seawater to effectively remove phenol and reduce its entry into condensed freshwater. The purposes of this study are: (1) to develop a new process for inhibiting phenol from entering condensed freshwater; and (2) to explore the contribution and mechanism of inhibiting the entry of phenol into condensed freshwater by activated PS during solar distillation.

## 2. Experimental Section

### 2.1. Materials

All chemicals purchased were of analytical grade and used as received without any further purification. Phenol, potassium monopersulfate triple salt (KMPS, 47% KHSO_5_ (PMS) basis), sodium molybdate, thiourea, and sodium sulfite were purchased from Aladdin Industries, Shenzhen, China. Sodium chloride (NaCl, 99.5%), ammonium sulfate ((NH_4_)_2_SO_4_, 99%), sulfuric acid (H_2_SO_4_, 95.0–98.0%), sodium hydroxide (NaOH, 96%), and methanol (99.5%) were purchased from the China Medicines Group of Companies, Beijing, China. Potassium persulfate (PDS, 99.9%) was purchased from Shanghai McLean Biochemical Technology Co., Ltd., Shanghai, China. Expandable polyethylene (EPE) foam was obtained from Shanghai Wuyou Packaging Co., Ltd., Shanghai, China. Polytetrafluoroethylene (PTFE) foam was purchased from Shenzhen Changxinda Plastic Products Co., Ltd., Shenzhen, China. Fiberglass membrane (FB) was purchased from Millipore Sigma, Billerica, MA, USA.

### 2.2. Preparation of the EPE/PTFE/FB/MoS_2_ Evaporator

The preparation of the antioxidative solar evaporator was divided into the following steps (Figure 1): First, MoS_2_ was prepared using a hydrothermal method [29]. 0.4 g of thiourea and 0.3 g of sodium molybdate were dissolved in 30 mL of ultrapure water under magnetic stirring, and the pH of the solution was adjusted to 1.0. The above solution was transferred to a stainless steel autoclave and heated in an oven at 200 °C for 12 h, then cooled to room temperature naturally. The MoS_2_ was collected and washed with ultra-pure water several times, dried at 60 °C for 12 h, and ground into MoS_2_ powder. Second, MoS_2_ powder was dispersed in 40 mL ultrapure water through ultrasonication for 30 min, followed by loading onto FB to prepare the photothermal material layer via vacuum filtration. Third, the MoS_2_ loaded FB was placed on top of the PTFE plate with a thickness of 1 cm. Fourth, the FB was also used as the seawater delivery channel and through the center hole of the EPE foam and PTFE. Finally, the antioxidation solar evaporator (denoted as EPE/PTFE/FB/MoS_2_ evaporator) was fabricated by combing the objects from step 3 and step 4.

### 2.3. Solar Evaporation

All solar evaporation experiments were performed in the laboratory at a temperature of 28 ± 1 °C and a humidity of 65 ± 10%. The EPE/PTFE/FB/MoS_2_ evaporator was placed into a cylindrical glass filled with 50 mL of 3.5 wt% NaCl solution (see Appendix A). During the process of solar evaporation, the weight change was recorded by an electronic balance (PTX-5102) purchased from Huazhi (Fujian) Electronic Technology Company, and a xenon lamp (XES-40S3-TT) purchased from SAN-EI Electric Japan with an AM 1.5 filter was used as a solar simulator. The irradiance intensity of 1 kW/m^2^ was defined as 1 sun. The average temperature distribution of the evaporator surface at 0 min, 5 min, 10 min, 30 min, and 60 min was recorded by taking thermal images with an infrared thermal imager (S65, FLIR).

### 2.4. Solar Distillation

First, a phenol solution (3.5 wt% NaCl solution or 3.5 wt% NaBr solution) with a concentration of 1 mg/L was prepared. Second, 50 mL of the above phenol solution was added to the cylindrical center of the condense device reported in our previous work [30]. Third, the EPE/PTFE/FB/MoS_2_ evaporator was placed in a phenol solution. Finally, the condense device was sealed with plastic wrap and placed under the xenon lamp (XES-40S3-TT, SAN-EI ELECTRIC JAPAN) with an AM 1.5 filter with an irradiation intensity of 1 kW/m^2^. The concentrations of phenol were analyzed by a high-performance liquid chromatography device (HPLC, Agilent 1200) using a separation column (Agilent Eclipse XDB-C18) (4.6 mm × 250 mm, 5 μm) and a fluorescence detector; the mobile phase was 50% water and 50% methanol with a flow rate of 0.8 mL/min, and the emission and excitation wavelengths were 275 and 313 nm, respectively.

### 2.5. Total Organic Chlorine and Total Organic Bromine Detection

A detailed total organic chlorine (TOCl) detection method was reported in our previous work [31]. Typically, the condensed freshwater was oxidized by the UV/O_3_ process for 60 min to ensure that the organic compounds were completely mineralized. The ozone gas concentration was 10 mg/min with the gas flow rate of 0.3 L/min. A 10W UV lamp (GPH212T5L/4, Hanau, Germany) with a wavelength of 254 nm was used as the UV radiation source. The TOCl concentration was recorded as the sum of inorganic chloride ions in the oxidized water. The sum of chloride ions, including Cl^−^, ClO_2_^−^, ClO_3_^−^, ClO_4_^−^ ions were determined using an ion chromatograph (SeoulDX120, Dionex, Sunnyvale, CA, USA). The method for measuring total organic bromine (TOBr) was the same as that for measuring TOCl, except that total bromide ions were tested instead of total chloride ions.

### 2.6. Distillation By-Products Detection

For distillation by-product detection, ^13^C-phenols with a concentration of 10 mg/L in different solutions (e.g., 3.5 wt% NaCl solution, 3.5 wt% NaBr solution, 3.5 wt% NaCl mixed with 0.09 wt% NaBr solution, and seawater from the East China Sea) were used as the target pollutants. The concentrations of the PMS and PDS were set at 1 mmol/L and 10 mmol/L respectively. After solar distillation, condensed freshwater was collected to test the total amount and species of halogenated by-products. Total organic carbon (TOC) was measured using a TOC analyzer (TOC-VCPN, Shimadzu, Kyoto, Japan). The halogenated by-products were detected by ultra-performance liquid chromatography/electrospray ionization triple quadrupole mass spectrometer (UPLC/ESI-tqMS).

## 3. Experimental Section

### 3.1. Solar Evaporation

Water evaporation based on interfacial solar heating takes advantage over other water evaporation since it can localize the solar heat in the air–water interface to rapidly increase the interface temperature. The average surface temperature of the EPE/PTFE/FB/MoS_2_ evaporator quickly reached 39.5 °C in the first 10 min under 1 sun irradiance, which indicates the good light-to-heat conversion capability of the evaporator. Then, the average surface temperature gradually enhanced to 42.1 °C after 30 min of irradiance and almost remained unchanged in the following 30 min (Figure 2a). The water evaporation process can be fitted with the zero-order kinetic, and the water evaporation rate was calculated to 1.35 kg·m^2^/h (Figure 2b), which is about 5 times higher than that of the 3.5 wt% NaCl solution evaporation without evaporator. The values of solar evaporation rate and air–water interface temperature are within the range of most other reported work (Appendix A).

### 3.2. Solar Distillation by Activated Persulfate

Before solar distillation, the antioxidation property of the EPE/PTFE/FB/MoS_2_ evaporator was tested since PS is a strong oxidant that can corrode the evaporator. The TOC concentrations of the condensed freshwater with and without addition of PDS were 1.1 mg/L and 1 mg/L respectively under 4 sun solar irradiance with the initial PDS concentration of 10 mmol/L (Appendix A), which indicates that no other new organic compounds are oxidized and enter into condensed freshwater. Therefore, the EPE/PTFE/FB/MoS_2_ evaporator can resist the risk of being oxidized in the activation of persulfate. Furthermore, it confirms that the EPE/PTFE/FB/MoS_2_ evaporator will not cause interference during distillation by-product analysis. For solar distillation, phenol was selected as a typical VOC. Here, the concentration of pollutants in the condensed freshwater was evaluated using the “pollutant distillation concentration ratio (R_D_), Equation (1) [11]”.
R_D_ = C_D_/C_B_(1)
where C_D_ and C_B_ are the concentrations of the contaminants in condensed freshwater and the initial simulated seawater, respectively. The lower R_D_ value indicates a relatively lower concentration in condensed freshwater. The R_D_ values of phenol were 76.5%, 104.9%, and 85.7% in pure water, 3.5 wt% NaCl solution, and 3.5 wt% NaBr solution, respectively (Figure 3), which means that the phenol evaporates with water vapor and condenses into freshwater. With the addition of PMS, the R_D_ values of phenol reduced to 6.4%, 0.0%, and 0.0% in the pure water, 3.5 wt% NaCl solution, and 3.5 wt% NaBr solution, respectively, indicating no phenol in the freshwater when using salt solution as the feedwater. By contrast, the R_D_ values of phenol reduced to 44.6%, 62.6%, and 26.6% in the pure water, 3.5 wt% NaCl solution, and 3.5 wt% NaBr solution, respectively, with the addition of the PDS. Here, the PMS has a lower R_D_ value of phenol than that of the PDS both in NaCl solution and NaBr solution. Because PMS can directly react with Cl^−^ or Br^−^ to generate HClO/Cl_2_ or HBrO/Br_2_ (Equations (2) and (3)), while PDS cannot [32].
Cl^−^ + HSO_5_^−^ → HClO + SO_4_^2−^
(2)
2Cl^−^ + HSO_5_^−^ + H^+^ → SO_4_^2−^ + Cl_2_ + H_2_O(3)
SO_4_^•−^ + Cl^−^ → Cl^•^ + SO_4_^2−^(4)
Cl^•^ + Cl^−^ → Cl_2_^•−^(5)
Cl_2_^•−^ + Cl_2_^•−^ → 2 Cl^−^ + Cl_2_(6)
Cl^•^ + Cl^•^ → Cl_2_(7)
Cl_2_ + H_2_O → HClO + H + Cl^−^(8)

For optimization of the R_D_ value, parameters such as PS dose, salinity, light initial phenol concentration, and light intensity that affecting the R_D_ values of phenol were investigated, and the results are shown in Figure 4. All the R_D_ values of phenol were zero with the addition of PMS in the range of 0.1–10 mg/L (Figure 4a), which means that a small dose (such as 0.1 mg/L) of PMS can completely inhibit phenol from entering condensed freshwater. The R_D_ values of phenol gradually decreased from 86.3% to 18.3% when increasing the PDS dose from 0.1 mmol/L to 5 mmol/L, then the R_D_ value of phenol reached 0 when further increasing the PDS dosage to 10 mmol/L. The effects of salinity on the R_D_ values of phenol were investigated by tuning the NaCl concentration range from 1.5 wt% to 10 wt% while keeping other conditions unchanged. The phenol in condensed freshwater could be completely removed with the addition of PMS at NaCl concentrations ranging from 1.5 wt% to 10 wt% (Figure 4b). However, with the addition of PDS, the R_D_ value of phenol increased with enhanced salinity. This is because the main factors for the activation of PDS are light and heat, and the halide ions have no effect on their activation efficiency. Conversely, the increased salinity led to an increased Henry’s coefficient of phenol, which causes phenol to be more likely to enter condensed freshwater [33]. The R_D_ values of phenol increased with the increased initial phenol concentration (Figure 4c) due to the competition of the active species (e.g., SO_4_^•−^, active chlorine) by the increased phenol molecules. The R_D_ value of phenol increased with enhanced solar intensity with the addition of both PMS and PDS (Figure 4d). This is because as the solar intensity becomes higher, the temperature of the air–water interface continues to increase, which in the next step activates PS to generate more active species.

### 3.3. Activation Mechanism

It is believed that the existence of various influence factors, including light, heat, MoS_2_ and Cl^−^ for the activation of PS. To explore the contribution of each factor to the activation of PS, the removal rate of phenol was tested by adjusting each single variable while keeping the other variables unchanged. For the activation of PMS, the removal rates of phenol by light, heat, MoS_2_ and Cl^−^ and blank activated PMS were 18%, 13%, 19%, 100%, and 19%, respectively (Figure 5), indicating that Cl^−^ is the main factor for the activation of PMS [34]. For the activation of PDS, the removal rates of phenol activated by light, heat, MoS_2_ and Cl^−^ and blank were 100%, 53%, 32%, 1%, and 4% (Figure 5), which indicates that light and heat are the main factors for PDS activation during interfacial solar distillation [35]. Therefore, it explains that the PMS has a lower R_D_ value of phenol than that of the PDS (see Figure 3). Moreover, it has been reported that Cl^−^ can be oxidized by SO_4_^•−^ to form active chlorine such as Cl^•^, Cl_2_^•−^, Cl_2_, and HClO [36] (Equations (4)–(8)), Br^−^ can undergo a transformation similar to Cl^−^ under the action of SO_4_^•−^ [37]. The reaction rate constant (k) of Br^−^ and SO_4_^•−^ is 3.5 × 10^9^/M·s [38], which is larger than that of the k (=2.1 × 10^8^/M·s [39]) value of Cl^−^ and SO_4_^•−^. Consequently, with the addition of PDS, the R_D_ value of phenol in the NaBr solution was lower than that in the NaCl solution (see Figure 3).

As with the formation of active chlorine and active bromine, halogenated distillation by-products may be formed during solar distillation when the seawater contains organic pollutants. To explore the total amount of halogenated distillation by-products, total organic halogen (TOX) was used as a comprehensive index to quantitatively analyze and compare different distillation by-products in condensed freshwater with the addition of PMS and PDS [40,41]. Under 1 sun irradiance, the concentrations of TOCl in condensed freshwater with the addition of PMS and PDS were 0.86 mg/L and 4.48 mg/L (Figure 6), which means a lower amount of chlorinated distillation by-products were formed with the addition of PMS. With the increase in solar intensity to 4 sun solar irradiance, the concentrations of TOCl in the condensed freshwater with the addition of PMS and PDS were 1.84 mg/L and 6.64 mg/L, respectively. This is because stronger solar intensity leads to higher temperatures on the evaporator surface. The stronger solar intensity and higher temperature are beneficial for the activation of PS to form more active chlorines, which in the next step produce more chlorinated distillation by-products. Moreover, total organic bromine (TOBr) with concentrations of 1.2 mg/L and 21 mg/L were found in condensed freshwater with the addition of PMS and PDS, respectively, suggesting the existence of brominated distillation by-products.

To further analyze the halogenated distillation by-products in condensed freshwater, ^13^C labeled phenols in different solutions (e.g., group 1: 3.5 wt% NaCl solution, group 2: 3.5 wt% NaBr solution, group 3: 3.5 wt% NaCl mixed with 0.09 wt% NaBr solution, and group 4: actual seawater) with concentrations of 10 mg/L were used as the feedwater, and the results are shown in Appendix A. With the addition of PMS, trichlorophenol, and monochlorophenol were detected in group 1, tribromophenol was detected in group 2, trichlorophenol, 2-bromo-2,5-cyclohexadien-1-one, 4-bromo-2,5-dichloropheno and tribromophenol were detected in group 3, and trichlorophenol, dibromophenol, dibromobenzoquinone, tribromophenol, and 2,4-dibromo-5-chlorophenol were detected in group 4. With the addition of PDS, trichlorophenol was detected in group 1, dibromophenol, dibromobenzoquinone, and tribromophenol were detected in group 2, and tribromophenol was detected in group 3, 2,4-dichlorophenol, 2,6-dichloro-1,4-benzoquinone, trichlorophenol, 2,6-dibromophenol, 2,6-dibromo-1,4-benzoquinone, 2-chloro-4,6-dibromophenol, and 2-bromo-4,6-dichlorophenol were detected in group 4. Based on the above by-product detection, a possible transformation pathway of the phenol was speculated (Figure 7). First, the formed active halogen reacts with phenol to form halogenated phenols, including monohalogenated phenols, dihalogenated phenols, and trihalogenated phenols. Second, the halogenated phenols can be further oxidized by the ^•^OH and SO_4_^•−^ to form monohalogenated cyclohexadienone, dihalogenated cyclohexadienone, and trihalogenated cyclohexadienone [42]. Meanwhile, monohalogenated cyclohexadienone can be further halogenated to dihalogenated cyclohexadienone and trihalogenated cyclohexadienone. Finally, the benzene ring is opened and transformed into a small molecular organic species.

## 4. Conclusions

In summary, we demonstrated the inhibition of phenol from entering condensed freshwater by activated persulfate during solar-driven seawater desalination. The activated persulfate process can make full use of the advantages of interfacial solar distillation, such as high surface temperature, sufficient solar irradiance, photothermal material, and high salinity, thereby effectively inhibiting phenol from entering freshwater. Compared with PDS, PMS had a higher phenol removal efficiency, lower dosage, and lower halogenated distillation by-products, which make it more suitable for inhibiting phenol from entering freshwater. More importantly, the problem that the formation of halogenated distillation by-products in condensed freshwater during the solar-driven seawater distillation process by activated persulfate should be given more attention in the future.

## Figures and Tables

**Figure 1 molecules-27-07160-f001:**
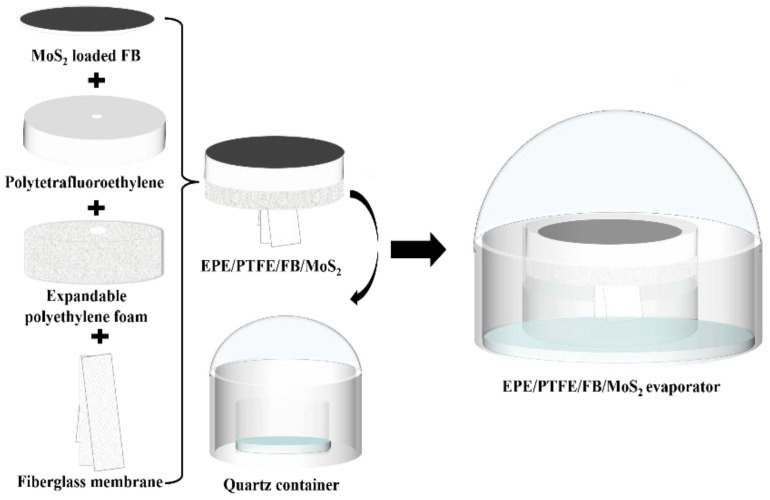
Schematic image of the fabrication process of the EPE/PTFE/FB/MoS_2_ evaporator.

**Figure 2 molecules-27-07160-f002:**
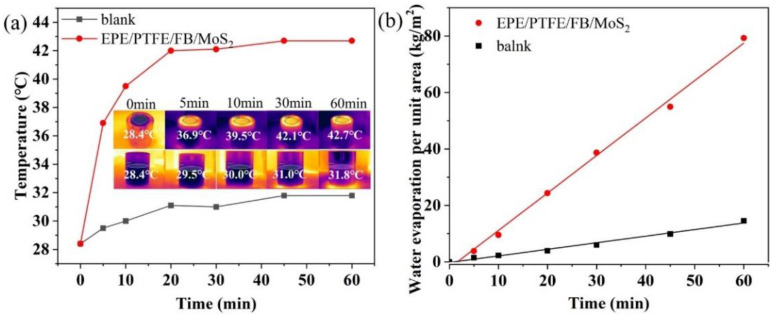
(**a**) The air–water interface temperature and (**b**) the mass of the evaporated water during the solar evaporation process of the 3.5 wt% NaCl solution with and without the EPE/PTFE/FB/MoS_2_ evaporator under 1 sun solar irradiance. Inset in **a**: IR images (**top**) with and (**bottom**) without the evaporator at different times.

**Figure 3 molecules-27-07160-f003:**
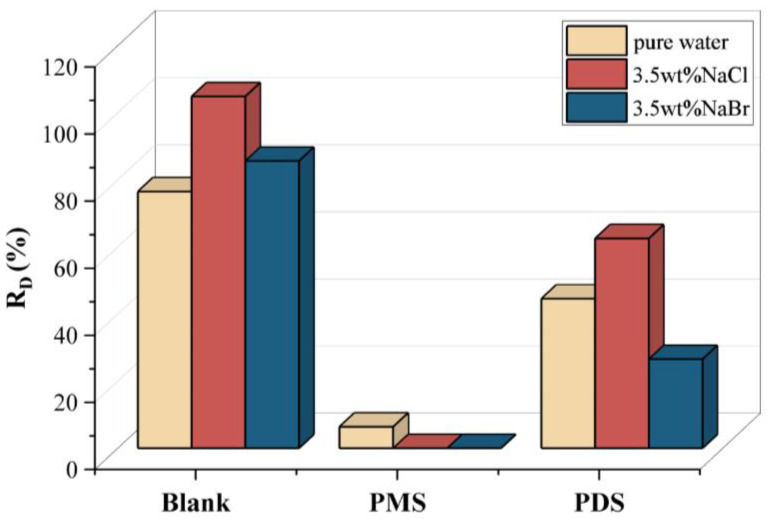
The R_D_ values of phenol with and without the addition of PMS and PMS in pure water, 3.5 wt% NaCl solution, and 3.5 wt% NaBr solution. Experimental conditions: [Phenol]_0_ = 1 mg/L, solar intensity = 1 kW/m^2^, [PMS] = 2 mmol/L, [PDS] = 2 mmol/L.

**Figure 4 molecules-27-07160-f004:**
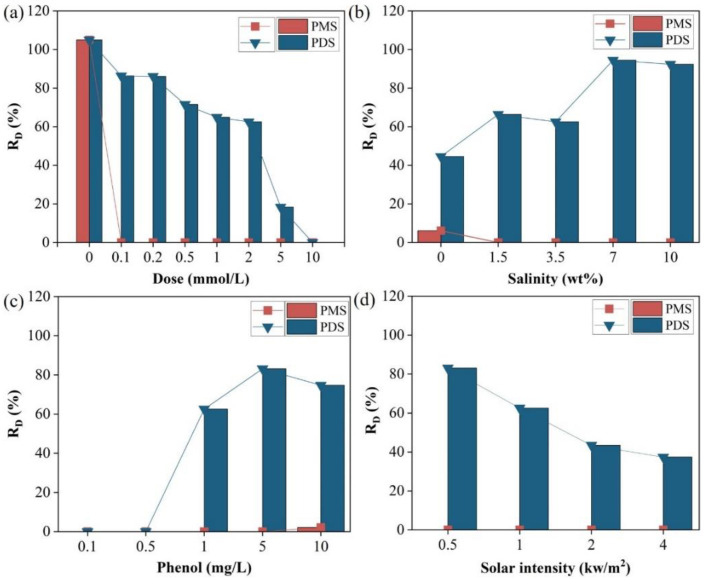
Effects of (**a**) PS dose, (**b**) salinity, (**c**) initial phenol concentration, and (**d**) solar intensity on the R_D_ values of phenol. Experimental conditions: [Phenol]_0_ = 1 mg/L, solar intensity = 1 kW/m^2^, [salinity] = 3.5 wt% NaCl, [PMS] = 2 mmol/L, [PDS] = 2 mmol/L.

**Figure 5 molecules-27-07160-f005:**
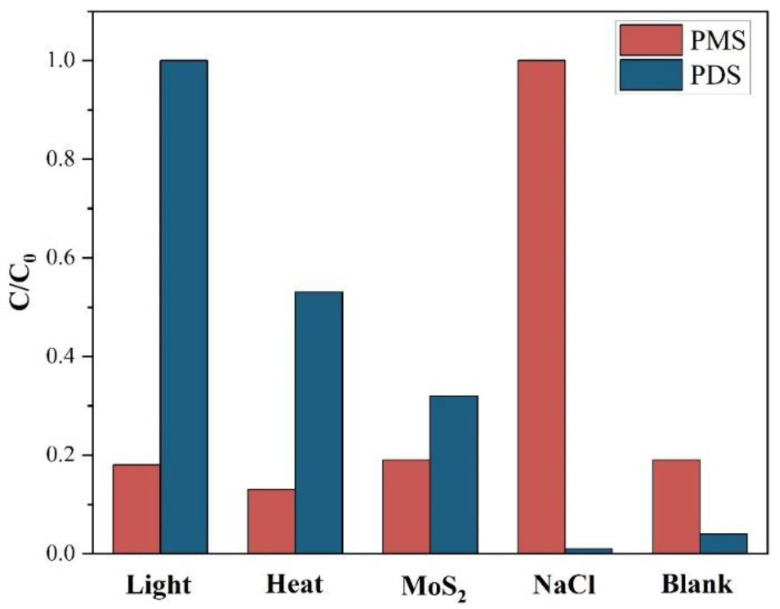
Contribution of each factor to the activation of PMS and PDS. Experimental conditions: [Phenol]_0_ = 1 mg/L, [PMS] = 1 mmol/L, [PDS] = 1 mmol/L, solar intensity = 1 kW/m^2^, T = 42.2 °C, [salinity] = 3.5 wt% NaCl, [MoS_2_] = 0.88 g/L.

**Figure 6 molecules-27-07160-f006:**
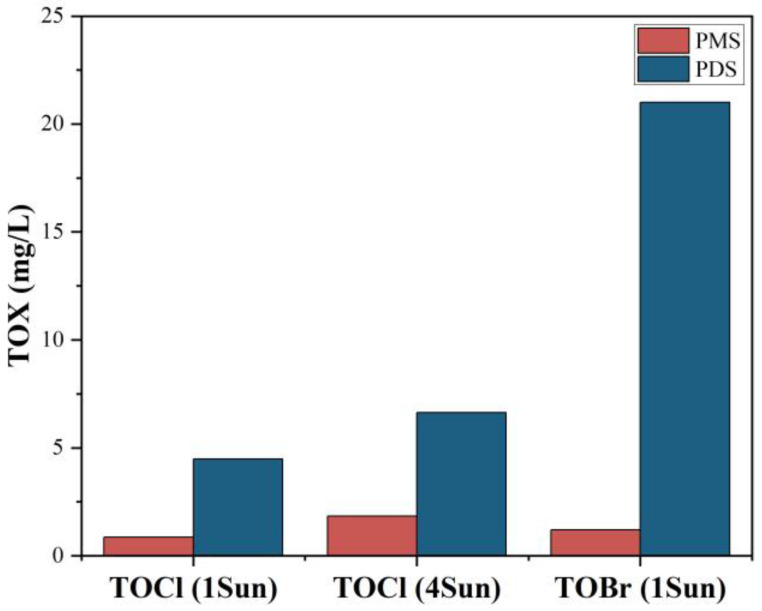
Total organic halogen (TOX) concentration in condensed freshwater with the addition of PMS and PDS under different solar intensities. Experimental conditions: [Phenol]_0_ = 10 mg/L, [PMS] = 1 mmol/L, [PDS] = 10 mmol/L.

**Figure 7 molecules-27-07160-f007:**
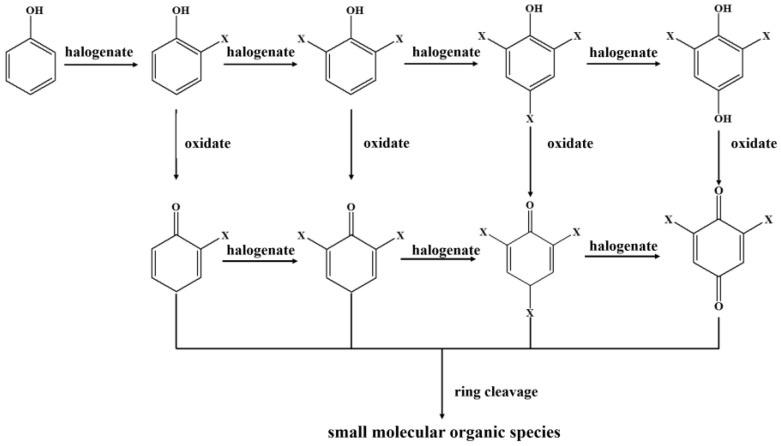
Possible transformation pathways of phenol during solar-driven seawater desalination by activated persulfate (X represents chlorine or bromine).

## Data Availability

Not applicable.

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
