# Peer review of "Inhibition of Phenol from Entering into Condensed Freshwater by Activated Persulfate during Solar-Driven Seawater Desalination"

_molecules, 2022, doi:10.3390/molecules27217160_

Round 1

Reviewer 1 Report

1. My major concern is the novelty statement of this paper. The authors are expected to demonstrate more on their new findings, with the statement starting with "for the first time", especially the general adaptivity. The general readers of this journal may be confused that this paper is just a testing report. 

2. The authors can add the secondary axis in some figures, to make the plots more readable. Currently, some histogram are hard to be recognized and the changing trend can not be represented well. 

3. The authors can attract more audience with more interesting background introduction at the beginning, with regards to the importance of sea water desalination, for example, to reduce scaling in the injection pipeline "https://doi.org/10.2516/ogst/2020045".

4. As always, a nomenclature table is recommended to better illustrate the meanings of the many abbreviations. 

5. The authors can comment on the economic considerations that may affect the practical adaptability of their optimization suggestions, or further demonstrate the superiority for industrial applications. 

Author Response

The reply to the reviewer #1 can be found in the attached file.

Reviewer 2 Report

The authors in this paper investigate the inhibition of phenol from entering condensed freshwater during solar desalination. These are my comments:

1- The results in the abstract are ambiguous and not that clear. It does not provide clear information to the reader.

2- The authors need to restate the "problem" and their contribution in the introduction section a bit more clearly. The authors briefly stated that VOC's can create a safety risk on condensed freshwater but failed to state the VOC's health/environmental hazards. There something wrong with the flow of information here.

3- Do you have actual pictures of the used solar evaporator? You can put it in section 2.3. I feel that it would give the reader a good perspective of the experiment.

4- In figure 2, the numbers and the thermal image should be larger in size. I can barely read the numbers on the axes.

5- How did you read the values of phenol in the water? 

6- In line 192-193: "small dose of PMS...". presenting this information like this can confuse the reader. Please quantify.

7- Figure 4: same goes here as well. Text and numbers should be larger in size.

8- In line 200-201: "RD value of phenol increased with the enhanced salinity" could you elaborate?

9- 257-261: "To further analysis the halogenated distillation...". It is a run-on sentence and does not make sense. Please re-write.

10- Supported materials provided little information to me. No explanation as to how/where this information could be beneficial in the paper.

Author Response

The reply to the reviewer #2 can be found in the attached file.
